# A Systematic Survey of Characteristic Features of Yeast Cell Death Triggered by External Factors

**DOI:** 10.3390/jof7110886

**Published:** 2021-10-20

**Authors:** Erika V. Grosfeld, Victoria A. Bidiuk, Olga V. Mitkevich, Eslam S. M. O. Ghazy, Vitaliy V. Kushnirov, Alexander I. Alexandrov

**Affiliations:** 1Moscow Institute of Physics and Technology, 9 Institutskiy per, Dolgoprudny, 141700 Moscow, Russia; grosfeld.ev@phystech.edu; 2Federal Research Center of Biotechnology of the RAS, Bach Institute of Biochemistry, 119071 Moscow, Russia; victoria.bidiuk@gmail.com (V.A.B.); mitkevich@inbi.ras.ru (O.V.M.); 1072195050@rudn.ru (E.S.M.O.G.); vvkushnirov@gmail.com (V.V.K.); 3Institute of Biochemical Technology and Nanotechnology, Peoples’ Friendship University of Russia (RUDN), 6 Miklukho-Maklaya Street, 117198 Moscow, Russia; 4Department of Microbiology, Faculty of Pharmacy, Tanta University, Tanta 31111, Egypt

**Keywords:** death, yeast, necrosis, apoptosis, autophagy, viability, vitality, ROS, caspase, permeability

## Abstract

Cell death in response to distinct stimuli can manifest different morphological traits. It also depends on various cell death signaling pathways, extensively characterized in higher eukaryotes but less so in microorganisms. The study of cell death in yeast, and specifically *Saccharomyces cerevisiae*, can potentially be productive for understanding cell death, since numerous killing stimuli have been characterized for this organism. Here, we systematized the literature on external treatments that kill yeast, and which contains at least minimal data on cell death mechanisms. Data from 707 papers from the 7000 obtained using keyword searches were used to create a reference table for filtering types of cell death according to commonly assayed parameters. This table provides a resource for orientation within the literature; however, it also highlights that the common view of similarity between non-necrotic death in yeast and apoptosis in mammals has not provided sufficient progress to create a clear classification of cell death types. Differences in experimental setups also prevent direct comparison between different stimuli. Thus, side-by-side comparisons of various cell death-inducing stimuli under comparable conditions using existing and novel markers that can differentiate between types of cell death seem like a promising direction for future studies.

## 1. Introduction

Cell death is a non-reversible breakdown of the normal functions of a cell that is accompanied by division arrest and loss of homeostasis. Cells constantly die during normal physiological processes such as development and differentiation-based cell turnover. This subset of cell death is termed programmed cell death. Cells can also die in response to various external stressors such as pathogens, toxins, environmental changes, or internal perturbations including mutations or epigenetic malfunctions. The established view is that if cell death involves activation or inactivation of some signaling pathways, it can be at least partially prevented using genetic or pharmacological perturbations; thus, such types of death are considered to be regulated cell death (RCD) as opposed to accidental cell death, which are unavoidable [1].

Cell death has been studied in all the major taxa of living organisms. In some cases, cell death involves specific machinery that has, at least in higher eukaryotes, presumably evolved for this specific purpose, among others. This includes such proteins as caspases, other pro- and anti-apoptotic factors as well as proteins allowing controlled formation of pores in membranes. Interestingly, homologs of proteins involved in cell death in higher eukaryotes are also present in single-celled eu- and prokaryotes (reviewed in [2,3]).

Higher eukaryotes have been characterized most extensively in terms of the different types of cell death mechanisms, and readers can review the current state of the field in the following work [1]. Although this review proposes at least 12 types of regulated cell death in higher eukaryotes, they are still commonly separated into the broader apoptosis, necrosis, and lysosome- and autophagy-mediated cell death categories. These are mostly distinguished by morphological and some biochemical criteria, while a more detailed determination of the cell death mechanism is often based on the specific machinery involved. In essence, apoptotic death is an active process that does not involve immediate permeabilization of the plasma membrane (PM) and the primary cause of death is often thought to be damage to proteins and DNA caused by activation of caspases and mitochondrial dysfunction. Necrosis involves breakdown of the PM barrier function as the primary cause of death. It can be both accidental as well as active when the process is regulated (reviewed in [4,5,6]). Lysosome-mediated death involves release of proteolytic enzymes from lysosomal compartments, which then cause general breakdown of cellular proteins, while autophagy-mediated death involves destruction of the cellular organelles and other components inside autophagic vesicles (reviewed in [1]).

Surprisingly, many of the features characteristic of apoptotic death in higher eukaryotes can be observed in single-celled organisms such as yeast [7]. These include condensation of chromatin, fragmentation of DNA, and exposure of phosphatidylserine. Later studies also identified the yeast metacaspase Yca1 [8], which had strong effects on cell death induced by hydrogen peroxide and chronological aging, among other stimuli. Other proteins that are homologous to members of higher-eukaryotic cell death pathways are much more ancient than multicellular organisms [9], although it seems that machinery related to cell death is not as developed in single-celled organisms as it is in higher eukaryotes. Despite this reduced form, the pathways responsible for cell death in simple organisms, such as yeast or bacteria, are less well explored and understood compared to higher eukaryotes. Notably, understanding cell death pathways in simple eu- and prokaryotes may have significant importance for the creation of drugs and other therapeutic methods aimed at treating the wide range of diseases caused by pathogenic microorganisms. Some researchers also suggest that bona fide programmed cell death, which is observed in both yeast and more complex fungi, could be harnessed as an antifungal strategy (reviewed in [10]. On a more basic level, altruistic PCD is highly likely to evolve in organisms that mostly live as clonal populations on spatially limited food sources such as yeast (see [11] for review) and, in fact, yeast have been shown to exhibit cell death behavior that can be viewed as altruistic [12,13].

The study of cell death in yeast has demonstrated that they can die in distinct manners in response to various stimuli, and this death can be analogous to the necrotic (reviewed in ([14]), apoptotic (reviewed in [15]), and, more rarely, autophagic and lysosome-mediated types of cell death in higher eukaryotes. Of note, this simplistic classification is mostly useful for the structuring of available data and may not represent an objective likeness to mammalian cell death. There is an ongoing debate on whether the various types of death observed in yeast and, more widely, fungi, should be termed apoptotic or necrotic (see [16]), with some authors suggesting that the term apoptosis should be applied only to cell death that has a highly analogous signaling cascade, which seems not to be the case for yeast. However, for the purposes of this review we will use the terminology set forth by the consensus paper on yeast cell death nomenclature [17]. We do not consider this terminology to be ideal, and invite interested readers to acquaint themselves with other viewpoints [16] as well as our own suggestions in the discussion section.

Interestingly, even in yeast, a simple organism, which has been studied quite extensively, several central features of cell death are not understood at an organismal level. What are the most vulnerable systems in a cell and how do they interact to kill cells upon perturbations? Why and how do specific stimuli cause specific types of cell death, or in some cases several types of death? Does each type of damage kill cells in a unique manner, or do many types of damage converge upon universal death mechanisms, of which there is a limited number? Answering these questions is of central importance for understanding how a complex system can experience breakdown and requires a high level of integrated knowledge of cell death which is currently lacking in the literature. While experiment-based systematic work is required to address these questions directly, the available literature provides a large swath of data, which could, potentially, be used to generate novel hypotheses and aim potential systematic research efforts. However, this literature has not been grouped in a manner which is amenable to easy searching based on specific stimuli and the available data on the features of cell death observed in each case. This review supplies such a resource, performs preliminary analysis of the gathered data as well as provides a wide view of the different external factors that can kill yeast cells.

## 2. Methods

One of the main results and instruments of this review was a table of papers that characterize cell death in response to various perturbations. Because the number of relevant papers we chose to include was ~700, all of them could not be cited in the main body of the manuscript, for which we apologize to the authors. The main reference section only lists papers that are central to the points made, while all of the papers that we reviewed are present Appendix A. We envision the use of the generated table as a tool to rapidly orient researchers on the types of perturbations that cause specific types of cell death in yeast, as well as to provide a broad view of the different perturbations that can cause death.

In total, we included data collected from the full texts of 691 papers, with data on 686 different stimuli. The compiled types of data are listed in generalized form in Figure 1. More than 7000 papers were screened from the raw keyword-based search results that we gathered according to criteria, presented in the Appendix A. Some stimuli were tested in multiple papers, while some papers included multiple stimuli, but only approximately 20 stimuli had more than 5 entries (not all of which were from different papers). Thus, cell death in yeast has actually been studied quite widely in terms of stimuli, but only a few stimuli were more or less well characterized. Figure 2 depicts the main criteria which were tested in most of the papers that were included in our analysis.

## 3. Methodological Aspects of the Literature Studying Cell Death

### 3.1. Methods for Detection of Cell Death

Studying cell death entails a clear definition of what dead cells are. However, this is not as trivial as it seems. For our purposes, a treatment induces cell death, if cells cannot resume division after it, when placed in conditions under which they can normally divide. A theoretical caveat of this definition is the possibility of cells acquiring some form of division arrest that is non-reversible under certain conditions, such as that observed for dormant Mycobacteria [18]; however, these situations have not been described in yeast and can probably be ignored for now. Death of cells can be detected using various methods, with the most direct one being the numbers of colony forming units (CFUs) before and after the treatment. This method is most useful in situations where the treatment kills large portions of the cell population, because the reported standard deviation of the mean in control samples can often exceed 10% [19,20]. Importantly, cell growth assays on stressful media, which are commonly used in microbiological studies, do not inform users on whether cells are dying or whether they are just slowing or arresting division completely. It is likely that most stressors cause a specific mix of cell death and slowed/arrested division, but, to our knowledge, this has not been studied thoroughly at present. The extreme case (i.e., when division stops completely) is studied more commonly as evidenced by the distinction between fungistatic and fungicidal drugs. Often, it is assumed that if the ratio between the minimal fungicidal concentration and the minimal inhibitory concentration is less than four, a treatment is considered to be fungicidal (this statement is commonly noted in numerous papers, such as [21], but, unfortunately, we could not find the original reference).

Another relatively reliable criterion for detecting cell death is an increase in cell membrane permeability to various molecules such as charged fluorescent probes. Although these probes have been reported to stain live cells under some situations involving cells’ high membrane potentials [22] or treated with anoxia [23,24], lipopolysaccharide [25], transient osmotic shock [26], or heat shock [27], most often cells with permeable membranes are indeed dead as evidenced by the good correlation between CFU counts and membrane-permeable populations in many scenarios such as treatment with hydrogen peroxide- and glucose-induced cell death performed in yeast [28,29].

Methods for detecting membrane integrity have the benefit of being much more sensitive to the identification of small populations of dead cells compared to colony counting methods. The most common approaches for assaying membrane permeability are the fluorescent dyes propidium iodide (PI) and Sytox Green (which stain nucleic acids), phloxine B (which, presumably, stains positively charged proteins), and methylene blue, which can be a membrane permeability dye or a vitality dye depending on the specific protocol (Figure 3). It is also known to bind DNA in vitro; however, whether this property is involved in the cell staining procedures is unclear [30]. Another stain recently reported to enter dying cells earlier than any noticeable PI staining is PO-PRO-1 [31], presumably due to the smaller size of the used molecule. Notably, some of these stains, namely, phloxine B and methylene blue, seem to work differently from what is often noted in papers (Figure 3). Phloxine B has been suggested to permeate both living and dead cells and then be excluded from living cells [32]. However, this view seems to be erroneous and not based on any experimental evidence. In our experience, phloxine B permeability is not dependent on drug efflux pumps (manuscript in preparation), and it is negatively charged in solution and should not enter cells with intact membranes. Since it is similar in structure to Eosin, it is also likely to bind to positively charged amino acids when inside cells. Methylene blue, which is commonly used to detect dead cells, is often thought to be enzymatically reduced from its blue form to a transparent one by living cells. However, in the most commonly used protocols based on acidic or neutral solutions, the dye is unable to enter living cells, thus acting in a similar way to PI, i.e., as a membrane permeability dye. It also does not experience any discoloration by living cell under these conditions. This can easily be verified by trying to monitor color changes of a cell suspension of interest. Methylene blue can be reduced and discolored by yeast in alkaline conditions [33], making it a vitality stain only under those conditions. However, this somewhat limits its use as a probe for cellular enzymatic activity under conditions where an alkaline pH is undesirable.

Another common approach is to monitor the release of compounds usually found inside the cell such as potassium ions, ATP, and proteins. Notably, release of potassium ions and membrane permeability to PI are not always in perfect correlation, probably because some perturbations only affect ion permeability but do not form openings large enough for more bulky molecules [34,35]. In addition, leakage of cellular material (phosphate and UV-absorbing material) caused by nystatin may not be correlated with cell death, although it is possible that extensive leakage could be detected only via the lysis of a small cell population, while a large fraction of cells did not leak [36].

Among high-throughput methods, luciferase-based detection of adenylate kinase leakage seems to be a highly convenient, [37,38] yet seldom used method. One must note that the papers that report its use do not provide an easy way of directly determining the share of permeabilized cells. Permeabilization of the membrane via different mechanisms might also release protein with different efficiency, so quantitative differences between different stimuli might not be easily interpretable.

A limited number of methods based on intracellular enzymatic activity have also been used in the literature as supportive methods for assaying the physiological activity of the cell, commonly termed vitality, with the most commonly used stains being methylene blue (see Figure 3), resazurin, FUN1 (2-chloro-4-(2,3-dihydro-3-methyl-(benzo-1,3-thiazol-2-yl)-methylidene)-1-phenylquinolinium iodide) as well as FDA (fluorescein diacetate). These methods seem to be informative for low-throughput applications and for studying cells subjected to acute stress; however, they require more extensive characterization if they are to be used systematically for comparing multiple conditions or mutants.

### 3.2. Differential Hallmarks of Cell Death

When it is quite clear that a certain condition causes cell death, there is often a need to further elaborate the mode of cell death. The easiest initial way to do this is to assay the correlation between the numbers of dead cells, as assayed by CFU-based methods, and the number of cells with permeable membranes. A high correlation suggests mostly necrotic death, while a low correlation is the first sign of prevalent non-necrotic cell death. The criteria for further elaboration of cell death type have been extensively reviewed in other reports [17], and are primarily based on the premise that non-necrotic death in yeast resembles apoptosis in higher eukaryotes and, thus, exhibits its characteristic hallmarks. In yeast, the most commonly tested hallmarks are exposure of phosphatidylserine (PS) on the outer leaflet of the membrane (via fluorescent Annexin V labeling), detection of DNA fragmentation and degradation using several approaches such as the TUNEL assay, electrophoretic methods, and flow cytometric DNA quantification as well as monitoring chromatin condensation and nuclear morphology using DAPI and Hoechst stains and electron microscopy. Another commonly assayed feature of non-necrotic yeast cell death is the role of the yeast metacaspase Yca1, as well as a set of other proteins homologous to higher eukaryotic pro-apoptotic proteins. For the sake of brevity, throughout the text we use the terms apoptosis and necrosis to address yeast cell death; however, the reader must bear in mind that most certainly the types of cell death initially termed apoptosis in mammals are not totally equivalent to the types of death observed in yeast. Thus, when we call a certain type of death apoptosis in yeast, we imply a set of methodological criteria and not necessarily complete or partial similarity to mammalian cell death. The relevance of some of these criteria to understanding cell death mechanisms in yeast will be discussed further.

Analysis of the aggregated data on cell death due to the action of different stimuli shows that clear necrosis with no apoptotic hallmarks was observed for 33 stimuli out of 209, for which these features were assayed. Clear apoptotic hallmarks with low prevalence of cell permeabilization were observed for 50 stimuli. The 126 remaining cases were those where both apoptotic and necrotic features were observed (All the noted stimuli are listed in separate tabs in Appendix A).

The existence of non-permeabilizing apoptotic death as well as the combined presence of apoptosis and necrosis begs the question as to why some stimuli do not cause secondary necrosis (or cause it quite slowly) at such as low concentrations of hydrogen peroxide [8], cisplatin [39], and human lactoferrin [40]. To our knowledge, this has not yet been studied.

#### 3.2.1. Detection of PS Exposure

Fluorescent annexin V staining is the most commonly used method to test for apoptosis-like death in yeast. It involves detecting the exposure of phosphatidylserine on the outer membrane leaflet. The standard logic [17] is to consider Annexin+PI– cells early apoptotic, Annexin-PI+ cells necrotic, and Annexin+PI+ cells late apoptotic (i.e. the cells have experienced secondary necrosis after apoptotic death). However, if membrane permeabilization provides openings large enough for Annexin to enter inside the cell, cells could be Annexin+PI+ but possibly not have anything to do with yeast apoptosis. Our data compilation shows that, at least in some cases where death is mostly necrotic (Appendix A, tab1) and annexin staining was performed, the fraction of Annexin+PI+ cells is quite low, suggesting that not all permeable cells are easily stained with Annexin (8 stimuli).

It seems reasonable that a considerable Annexin+PI– population should be the best indicator of apoptotic death that was slow to progress to necrosis, while a large Annexin+PI+ population might be due to the rapid post-apoptosis secondary necrosis, or non-specific annexin staining. In this context, the Annexin+PI– population was at least 5-fold larger than the overall PI+ population for 31 stimuli (Appendix A, tab2) out of the 153 stimuli for which it was tested.

Importantly, PS exposure in higher eukaryotes acts as a signal for cell phagocytosis, and the process of its exposition is regulated. Yeast most likely do not engulf each other in any similar manner, and so some researchers doubt whether Annexin staining provides any useful insight on the manner of death in yeast. In our opinion, since PS is asymmetrically distributed between the membrane leaflets in both yeast and higher eukaryotes, Annexin staining (with the noted caveats) allows visualization of changes to the membrane structure during death, providing a useful parameter for differentiating between various cell death scenarios. The possibility that, in yeast, PS exposure has no biological function does not diminish its usefulness as a diagnostic feature. Notably, recent work suggests that phosphatidyl serine exposure can also occur during non-apoptotic death in mammals [41] and that this exposure might not involve the action of scramblases or other flipping enzymes, but can involve aberrant formation and exocytosis of vesicles [42,43,44]. Thus, the position that yeast have no specific enzyme for PS flipping to the outside leaflet is, to our mind, not highly relevant, because numerous mechanisms might be involved.

#### 3.2.2. Yeast Metacaspase

Since the discovery of the yeast metacaspase [8] and its role in yeast cell death, a large number of studies have assayed the role of this protein, most commonly by using fluorescent inhibitors (FITC-VAD-FMK) or, more rarely, fluorogenic substrates as well as downregulation of *YCA1* (*MCA1/PCA1* in some yeast and fungal species). Using one or several of these methods, the effects of YCA1 (or its analogs) were observed for 97 stimuli, and for 36, the death was caspase-independent (Figure 4).

It is important to note that FITC-VAD-FMK, the most commonly used stain for caspase activity in yeast, has been shown to bind dead permeabilized cells subjected to cell cycle arrest via the cdc13-1 mutation [45] or cells subjected to an aging colony environment [46]. However, for at least 18 stimuli for which both FITC-VAD-FMK staining and cell permeability were performed, FITC-VAD-FMK positive cells were two-fold more prevalent than PI-positive cells, and in nine papers this ratio was 5-fold or higher (Appendix A tab 3). Even though in most of these papers, the PI and FITC-VAD-FMK staining were performed in separate experiments, we feel this shows that not all of the data obtained with FITC-VAD-FMK is due to non-specific staining of dead cells. Another concern is the ability of FITC-VAD-FMK to stain cells with deleted Yca1, which has been observed (personal communication by D. Knorre) but not reported in print.

If we look only at apoptotic stimuli for which the role of Yca1 was assayed using a deletion strain, 33 stimuli caused Yca1-independent death (28 of these were deemed apoptotic), while for 29 the death was diminished by *YCA1* deletion (Appendix A, tabs 6 and 7).

In several cases the data obtained via gene deletion and caspase inhibitor were discordant, which suggests that some other protease might bind the substrate, as reported for treatment with the ER-stressor tunicamycin [47] and antifungal plant defensin RsAFP2 [48].

In summary, a modulatory role of Yca1 in cell death in response to different stimuli seems quite certain and common. We must, however, point out that apart from its role as a putative pro-apoptosis factor, Yca1 is known to be involved in general proteostasis [49,50,51]. Absence of Yca1 can increase the levels of various stress response proteins, thus preconditioning cells to be more resistant to various types of shocks. Yca1 deletion also perturbs cell cycle progression [52], which might lower the susceptibility of cells to various death stimuli. Thus, while Yca1-death dependence does exist, the interpretation that it is a clear indication of the apoptotic nature of cell death can be questioned. However, it can still be used as a parameter for the formal differentiation of different cell death types (i.e., those that are Yca1-dependent and -independent).

Unfortunately, no other gene thought to be involved in yeast apoptosis has been tested for so many stimuli. However, we did include available data on gene deletions (as well as other treatments) that were shown to reduce the extent of the observed cell death in response to various stimuli (Appendix A, preventive treatment column). It is our feeling that testing the effects of these genes on cell death in response to a wide panel of stimuli would be a clear step in providing a novel framework for understanding the connection between the cell death stimulus and death type in yeast and providing a more defined set of criteria for death types.

#### 3.2.3. Reduction of Cell Death by Cycloheximide Treatment

Another commonly assessed feature is the effect of cycloheximide treatment on cell survival under a death stimulus. It is often considered that the positive effects of cycloheximide are indicative of apoptotic death, since apoptotic death mechanisms are thought to require protein synthesis. An alternative explanation would suggest that the protective effects are not related to the prevented synthesis of specific pro-apoptotic proteins but rather to the ability of cycloheximide to slow down translation of the majority of proteins and, thus, possibly triggering a universal reaction to numerous types of stress (reviewed in [53]). This idea is highlighted by the results reported in [54], since cycloheximide treatment strongly reduced cell death if added 30 min prior to a moderate heat shock (45 °C) but not if added immediately prior to, or during, the shock, even though it is known to stop translation quite rapidly. Thus, some aspects of the response to cycloheximide, apart from translational arrest, require some time to develop. Importantly, cycloheximide can also have negative effects on cell death; for instance, pretreatment with the compound reduced the acquired thermotolerance of yeast at 50 °C which was achieved by preincubation at 37 °C [54]. Most likely, this was due to the prevented synthesis of Hsp104, as under similar conditions, deletion of Hsp104 is known to strongly reduce survival [55]. Thus, even though the mitigating effects of cycloheximide are often viewed as indications of apoptosis, the situation is more complex and the simplistic explanation that the absence of the synthesis of this novel protein is the only reason for the mitigating effects of CHX is questionable.

#### 3.2.4. Lysosome- and Autophagy-Mediated Cell Death

A much less studied aspect of yeast cell death is the involvement of vacuolar proteases and autophagy. On one hand, in the general study of cell death, autophagy-mediated death is often listed as a specific subtype of cell death (reviewed in [1]). However, there are also reports of the beneficial role of autophagy during death-inducing stress [56]. Only a small number of papers have attempted to address the role of the autophagic system in yeast cell death, and there is no clear consensus on which methods can be thought of as reliable for these purposes. Most commonly, researchers assess the autophagic flux by monitoring GFP-Atg8 or other reporters, analyze of the effect of autophagy-related gene deletion on cell death or use more indirect measures such as changes of vacuole morphology and mitochondrial fragmentation. At least some data on the effects of autophagy or vacuolar proteolysis on cell death are available for 37 stimuli, with some indication of changes to the lysosomal activity or autophagy available for 26 stimuli. However, in all cases, except for one [56], involving zinc toxicity, perturbation of the mentioned genes increased sensitivity to the death stimulus (Appendix A). Most of remainder exhibited some feature characteristic of changes to the function of the vacuole but did not test for a causative relationship in cell death. Further in the review we present several of these cases such as death during sporulation and fatty acid treatment [57,58]. We feel that a more systematic study of the role of autophagy and lysosomal compartments in cell death would be useful to either demonstrate the role of autophagic processes for a subset of stimuli or disprove it. At the current state of the field, there is very little data on the pro-death role of autophagy and lysosomal compartments in yeast.

#### 3.2.5. Oxidative Stress and Reactive Oxygen Species

Another commonly assayed feature of cell death is general oxidative stress, the reasons being that reactive oxygen species (ROS) often form during stress and are thought to act as a trigger for the initiation of RCD or as the direct cause of death-inducing damage. The types of ROS and ways in which they damage cells have been reviewed extensively [59]. Most commonly, oxidative stress is detected using fluorogenic substrates such as 2′,7′-dichlorodihydrofluorescein diacetate (H2DCFDA), dihydroethidium, and dihydrorhodamine 123. These stains are sometimes thought to have specificity towards different ROS. However, this has been the subject of strong criticism (see [60] for a review). Another approach, which also has the benefit of determining a causative relationship between ROS and cell death, is assaying the effect of antioxidants on cell survival. However, this approach is also imperfect, because different antioxidants can have dramatically different effects, and there is no standard and agreed-upon panel of antioxidants for this purpose.

Overall, 224 distinct death-inducing stimuli were tested for ROS formation, and 13 showed no signs of ROS formation, while 217 did (Appendix A). Notably, this analysis is based on data from single papers, and several (but not all) of the stimuli reported not to induce ROS formation by one paper were shown to do so by other authors, albeit sometimes the conditions were different. This is an important caveat of all the presented analyses, which were not aimed at forming a consensus picture for each individual stimulus.

ROS formation is sometimes considered to be a hallmark of apoptosis, and the vast majority of cases where apoptosis was observed do show signs of ROS formation. However, cases of exclusive necrosis where ROS generation was tested, show that 16 stimuli exhibited ROS generation, while only two did not (Appendix A, tabs 4 and 7). These data would suggest that ROS generation accompanies most cell death-inducing stimuli, however, confirmation bias might also be in play, since testing of ROS formation might not be reported in cases where none was seen.

While ROS often form during different types of cell death, the question of their causative role in cell death is not always addressed. One of the most direct ways to do this is to test the effects of anti- and pro-oxidants on the specific case of cell death. Another common, but indirect approach is to determine the role of mitochondria, which are a central source of ROS in yeast, by testing the effect of respiration incompetent mutants or respiration-blocking drugs. If analyzing all of these approaches in aggregate, 93 stimuli that induce ROS formation and were tested for ROS causality in death in some manner, only nine showed no causality, while two showed reverse causation, i.e., antioxidants exacerbated cell death [61,62] (Appendix A, tabs 2 and 3). However, caution must be taken in conclusions regarding the overwhelming causative role of ROS, because a large number of stimuli that triggered ROS formation were not tested for causation. A reasonable assumption would be that due to the confirmation bias, the number of cases with non-causative formation of ROS in this category would be quite high. Figure 5 shows how ROS detection and causation are distributed among stimuli separated in the apoptotic, necrotic and apoptosis with necrosis categories. To sum up, a major problem for a systems level understanding of the role of ROS in cell death is the difficulty of assaying the presence of all of the possible ROS and of determining their causative effects, due to the limitations of current methods. Possibly, more universal and informative methods for ROS detection should be based on the expression levels of genes or production/activities of proteins involved in the oxidative stress response, while agreed-upon panels of antioxidants or mutants tailored for wide coverage could be used to identify the causative role of ROS in death.

#### 3.2.6. Calcium Signaling

Calcium signaling is known to play an important role in apoptosis in higher eukaryotes, both in terms of membrane repair [63] and in pro-death signaling [64]. It is also known to directly activate yeast metacaspase [65]. However, the role of calcium in cell death is assayed much more rarely than that of ROS in yeast and fungi. Overall, 60 distinct stimuli were reported to involve some aspect related to calcium signaling; however, for most of these, the causative role of calcium in cell death was not tested (Appendix A). For eight stimuli, reduction of the extra- or intracellular levels of calcium could mitigate cell death. Use of EGTA or calcium-depleted medium prevented death in response to carbon source depletion [66], the intracellular calcium chelator BAPTA-AM prevented death in response to glutathione depletion [67]. Ruthenium red is thought to inhibit Ca^2+^ uptake by the mitochondria but possibly has a more complex effect on calcium transport [68]. This compound prevents death in response to several toxic flavone compounds [69,70]. Along with several other manipulations of calcium uptake and intracellular release, ruthenium red also prevented death of *C. albicans* in response to a peptide derived from human lactoferrin [71].

On the contrary, for pheromone toxicity in both *S. cerevisiae* and *C. albicans,* Ca^2+^-depleted medium [72,73,74] and genetic perturbation of calcium signaling pathways [73] exacerbated cell death. Calcineurin-mediated signaling, which requires a burst of cytosolic calcium, was also required to prevent fluconazole-mediated death [75]. Somewhat similarly, pre-treatment with non-fungicidal concentrations of amiodarone could reduce cell death in response to heat shock [76]; however, the effect of amiodarone is not necessarily mediated by Ca^2+^ influx.

Finally, although it is not clear whether this was causative to cell death, a burst of cytosolic calcium was observed after treatment with several toxic peptides [77,78,79].

In terms of the cell death types accompanied by calcium-related events, only two stimuli out of 29 involved pure necrosis (i.e., tunicamycin and an equisetin-like compound), with the remainder being either pure apoptotic death (10) or a mix of apoptosis and necrosis (17). Whether this was caused by selection bias or is a true feature of apoptotic death is unclear. In general, it is likely that calcium plays a role in a much wider range of stimuli than currently visible from existing data (Appendix A). We hope future research can address this by assaying the changes in calcium levels in response to a wider range of death stimuli as well as testing for calcium dependence more widely.

#### 3.2.7. Changes in the Functionality and Morphology of Organelles

Cell death is often accompanied by specific morphological changes in various organelles. Electron microscopy is widely used to assess the morphology of the whole cell. For example, it is possible to detect lipid droplets and lack of all organelles, which are observed during lipid-mediated and lysosome-mediated death, respectively [57]. Necrotic death is accompanied by visible membrane disruptions [80], and apoptotic death shows nuclear condensation and PM blebbing [81]. Less often, the PM potential is studied, and the most common method to monitor it is staining with fluorescent dye DiBAC4(3). Ionic gradients and selective permeability of the plasma membrane for charged molecules are central for the transport functions of the cell and its integrity; thus, testing for cell plasma membrane function could provide useful information for understanding the mechanism of cell death, especially in response to external treatment.

Mitochondrial dysfunction, such as the loss of mitochondrial membrane potential, is considered to be an early event of apoptosis [82]. Since mitochondria are also a major source of ROS [83] as well as of other apoptosis-related actors such as cytochrome C [82], its role in cell death is commonly investigated by cell death researchers. Most often, this involves testing mitochondrial function by monitoring membrane potential with the use of fluorescent dyes (such as rhodamine 123, Mitotracker dyes, JC-1). Mitochondrial morphology (location in cell and mitophagic processes can also be detected using specific dyes or mitochondrial GFP fusion proteins. Importantly, complete lack of a mitochondrial genome or more specific disruption of mitochondrial functions via drugs reduces cell death in response to various factors such as alpha-factor, lactoferrin and killer toxin, as well as n-acetyl-sphingosine and free fatty acids [84,85,86,87,88,89,90,91], so either mitochondrial dysfunction in these cases is causative in toxicity or the absence of mitochondria contributes to the accumulation of some entity that prevents death.

#### 3.2.8. Role of Treatment Severity in Determining Cell Death Type

An important feature of cell death in response to various stimuli is that the same stimulus can cause different types of death, depending on the severity of the stimulus. This includes both concentration (for various chemical and biological entities), intensity of treatment (for physical treatments) and treatment duration. Several treatments, such as the archetypal treatment with hydrogen peroxide, have been shown to induce apoptosis-like death at lower concentrations, progressing to necrotic death with more severe treatments. We also noted the severity of treatments in our compilation, although, because of the large number of additional experimental differences, these data cannot be compared easily. This prevents the creation of a common comparative metric which could help collate cell death severity and stimulus intensity with markers of various type of cell death. In other words, comparing stimuli in terms of their “apoptotic/necrotic ranges” is not possible. In order to rectify this problem, we propose creating formal criteria, based on and similar to those of minimal inhibitory (MIC) and fungicidal concentration (MFC). For instance, percentages of necrotic and apoptotic cells at MIC, 50% lethality concentrations and MFC (with identical treatment times and medium) could be measured and compared for different stimuli.

### 3.3. Death-Inducing Perturbations in Yeast

While previous sections described the general outlines of how different stimuli trigger cell death in yeast and the main features used to study the cell death process, this section focuses on more specific groups of death-inducing stimuli as well as the kinds of death these stimuli induce. We have divided these stimuli into the following broad groups—stimuli related to the natural lifecycle of yeast (aging, mating, interaction with other species), shock of physical nature (heat, radiation, etc.), nutrient imbalance, acids, alcohols, peptides, and other biochemical stimuli not included in other categories.

#### 3.3.1. Yeast Life Cycle Related Stimuli

Yeast living in laboratory and natural environments experience a wide range of more or less physiological situations, i.e., where no clear stressor is being applied or where the changes in the environment seem similar to those that can be encountered during the normal life cycle of a yeast cell.

##### Replicative Aging

A single yeast mother cell can perform a limited number of divisions, after which it arrests and dies. The physiological changes of ageing cells and treatments that can extend replicative lifespan have been studied most extensively and have been reviewed numerous times; however, the specifics of cell death during aging have been characterized in only one paper which reported oxidative stress, annexin staining in the absence of widespread membrane permeabilization, and DNA fragmentation [92]. Replicatively ageing cells seem to experience increased length of cell cycle phases in the several divisions prior to death [93], after which they cease dividing and experience either quick lysis (probably corresponding to necrosis) or can exist in a non-necrotic state for hours. Recently, it has been demonstrated that one of the reasons for the death of aging cells is incorrect segregation of the chromosomes during division, which is usually corrected before the division is completed but can often result in death of an aged cell [94]. Whether or not this process produces apoptotic hallmarks in either the mother or daughter cell is currently unclear.

##### Chronological Aging

Another mode of aging, termed chronological ageing, is in actuality the process of slow cell death in a medium, where cells have depleted the nutrients required for growth and/or accumulate toxic metabolites. Most of the work done on chronological aging uses depleted synthetic medium with glucose as the sole carbon source. In these conditions, cells exhibit gradual death with time which involves hallmarks of apoptotic death such as DNA fragmentation, chromatin condensation, and oxidative stress [95]. Under conditions of chronological aging in rich medium, cells that do not acquire a quiescent state [96] also seem to experience both apoptotic and necrotic death, judging by an increased population of TUNEL positive cells and Annexin V staining cells as well as PI-positive cells without other apoptotic markers. In general, death during chronological aging has been studied quite extensively and has been reviewed in several papers [97,98].

##### Colony Aging

Another type of cell death is observed in colonies of yeast. For colonies aging on medium with glycerol as the sole carbon source, cells on the outside and inside of the colony display morphological differentiation, and cells in the colony centers show increased death. This was shown to be beneficial for cells located on the periphery of the colony, presumably via nutrient release. Thus, this type of death is one of the few that can truly be considered programmed death according to the consensus recommendations of yeast cell death researchers [17]. Notably, this type of cell death is independent of Yca1 and instead is controlled by ammonia signaling [12]. Colonies growing on synthetic medium with glucose as a carbon source also seem to experience cell death in the colony center that involves formation of toxic ROS, and this death has apoptotic features [86]. Yca1 dependence of this type of cell death was not tested. Since no ammonia signaling should occur in the colonies used in the latter work due to the young age of the colonies, the two types of cell death are likely to be distinct, although this was not tested directly.

##### Sporulation

Generation of spores is also a process that involves programmed cell death in yeast. It is quite common in conditions of low-carbon medium sporulation that only 1–3 of the 4 spores are viable, while the others die. This death has been shown to have characteristics of both apoptosis (DNA fragmentation) as well as lysosome- and autophagy-mediated death, since the vacuole of the cell is ruptured and the nucleus of the abortive spore is degraded in a process similar to macroautophagy [99]. Upon completion of spore generation, the mother cell and its organelles also experience cell death that combines characteristics of necrosis and autophagy [58].

##### Mating Associated Death

Action of the alpha-factor, a yeast-mating pheromone, has also been implicated in inducing cell death. According to one report, using wild-type yeast and high pheromone concentrations, cells experience apoptotic death [85]. Another report, using lower concentrations of pheromone in wild-type strains lacking a secretable protease that degrades the pheromone, showed that cells seemed to die necrotically in a calcineurin/Fig1- dependent manner. However, inhibition of this type of death uncovered multiple slower waves of death with different properties [73]. Interestingly, there are reports that pheromone-induced death seems to be variable both among different strains of *S. cerevisiae* [100], as well as isolates of *Candida albicans* [74]. Thus, although this type of death is present in different yeast genera, it does not seem to be highly conserved.

##### Interspecies Death Induction

Numerous secreted killer toxins are produced by various yeast species, and these can kill sensitive yeast strains. These proteinaceous entities differ in their mechanisms of action. However, most of them show at least some degree of necrotic death, while a subset has been shown to cause death with apoptotic features [91,101,102]. In general, killer toxins have various targets, with many of them targeting known or unknown entities on the cell wall. Subsequently, some toxins lyse cells through their effects on the cell wall [103], or interact with the cell membrane [104]. Others seem to have intracellular targets, and are more commonly associated with apoptosis-like death [91,101,102]. A more focused review of a subset of killer toxins is available for the interested reader [105].

Another fascinating instance of cell death observed in yeast is contact-mediated death upon exposure to other yeast species. This was first observed in a paper studying co-culture of *S. cerevisiae* and two sensitive yeast species—*Kluyveromyces thermotolerans* and *Torulaspora delbrueckii* [106]. Unexpectedly, this cell death required direct physical contact between cells. Although the type of cell death was not characterized extensively, it was resistant to cycloheximide treatment, which may suggest that it was non-apoptotic. A later paper studying a similar phenomenon on other yeast species *(Hanseniaspora guilliermondii,* and *Lachancea thermotolerans)* cultivated together with *S. cerevisiae*, demonstrated that *S. cerevisiae* accumulated GAPDH-derived peptides on its cell surface, which seemed to mediate the contact-induced death [107]. It would be interesting to understand which kind of cell death could be induced by peptides present on the cell surface, because this should most probably be a signaling-mediated effect.

Death via interaction between yeast and other fungal or bacterial species through secreted entities have also been studied [81,108,109]. However, the amount of data is too small to identify any trends, especially since the active entity is not quite clear.

Yeasts are also susceptible to infection by viruses, although quite commonly strains of yeast adapt to the virus, making it relatively benign (see [110] for a review). However, sensitive cells can die via damage to the cell membrane or via an apoptosis-like death [101,111].

A more exotic instance of cell death induced by inter-species interaction is contact-mediated rupture of yeast cells on the nano-structured surfaces of insects [112]. This paper demonstrates that the surfaces of wings obtained from several insects contain nano-scale patterning of their surface which can cause necrotic death of yeast cells upon adhesion. This effect can be observed using both intact wings as well as those covered by a ~10 nm layer of gold which negates contact chemistry, suggesting that the mere shape of the surface is sufficient to have a killing effect. A similar effect was also observed on artificial nano-structured surfaces [112].

#### 3.3.2. Shocks of Physical Nature

Numerous physical treatments can kill cells, and some of these stimuli are physiological for yeast, which can confront them in a natural habitat: temperature perturbations (heat and cold shock, gradual temperature change or periodic temperature oscillations), hyper- and hypoosmotic shocks, and different types of radiation (UV being the most common). There are also stimuli that yeast are unlikely to encounter—exotic types of radiation (gamma- and beta-), electro-magnetic fields, and various nanocomposite materials (nanotubes, nanoparticles, etc.). Some attempts were made to characterize the mechanisms of cell death caused by all of these types of physical stressors.

One of the most studied death-inducing physical stimuli is acute temperature shock, which is widely used for decontamination. Of course, severe heat shock causes immediate accidental death by complete breakdown of all cellular systems; however, death in response to milder shocks seems to proceed differently. It is dependent on ROS (most likely derived from mitochondrial dysfunction), and yeast cells could be partly rescued from heat shock-mediated death by antioxidants [113] or overexpression of cellular antioxidant system components [114] as well as by mitochondrial uncouplers [113]. Heat shock triggered cell death at 55 °C could also be mitigated by deleting the 14-3-3 protein Bmh1 [115], while death in response to gradual temperature increase could be lowered by a wide range of genes [116]. These facts suggest that, at least up to a certain intensity, heat shock can trigger regulated cell death. This idea is confirmed by the fact that human Bcl-xL partly protects yeast cells from heat shock-induced death [117].

Osmotic shock seems to cause both necrotic and apoptotic death due to inadequate or anomalous membrane reorganization. Importantly, hyperosmotic shock is deadly due to the presence of two factors, one of which is dehydration (hyperosmotic shock per se), while the other is subsequent rehydration (i.e., restoration of normal external pressure). Both of these cause noticeable permeabilization of the membrane [118,119], but the cell death is not completely explained by cell permeabilization in one of the reports [119], supporting the results of [120], which indicate the presence of apoptosis-like death based on both CFU counts and markers of apoptotic death (TUNEL and DAPI staining).

Ultraviolet radiation causes DNA modifications and produces a variety of different ROS in all cellular compartments (reviewed in [121]). The ROS, as we could estimate, play a crucial role in cell death caused by UV and this effect is strongly dependent on the dose of radiation [122]. In [123], the authors presume that the killing effect of UV radiation is realized via DNA damage and subsequent apoptosis is induced in this case based on chromatin condensation and reduction of DNA content. UV irradiation, in fact, causes a dose-dependent increase in annexin staining and caspase activation, both of which could be observed in non-permeable cells [124]. Primary necrosis has also been reported at high doses of UV irradiation [122].

Ionizing radiation (X-rays and gamma-rays) can produce ROS by water radiolysis [125] but can also make double-strand breaks in DNA molecules, induce base damage, and inactivate proteins. Antioxidants, such as N-acetyl cysteine, can reduce the oxidative cell damage caused by ionizing-radiation induced ROS [126] as well as protect against radiation-induced double-strand breaks and DNA deletions [127] but do not efficiently prevent cell death. There is no detectable PI staining during the first several hours after radiation treatment, which suggests that when necrosis is observed, it is likely secondary [128]. Ionizing radiation also induces some apoptotic events (such as caspase activation), but these events also seem to be secondary to DNA damage. Thus, we suggest that severe damage to DNA by ionizing radiation is what probably causes irreversible division arrest, while apoptotic and necrotic mechanisms and ROS might play only a secondary or minor role. This might argue in favor of the so-called mitotic catastrophe as a distinct type of cell death in yeast [129].

Nanomaterials are another type of physical stimulus that can induce cell death. The main types of materials studied in this respect are metal nanoparticles (gold, silver etc.) and carbon nanotubes. From the limited amount of published data we were able to compile, treatment with metal nanoparticles causes ROS-independent death (despite increasing overall ROS levels), mostly accompanied by a strong apoptotic phenotype [130,131].

#### 3.3.3. Nutrient Imbalance and Depletion

Several types of nutrient imbalances have been shown to cause death in yeast cells. The fact that yeast can die in response to these types of stimuli, often in a matter of hours, is surprising, because in pure water devoid of any nutrients, *S. cerevisiae* can survive for weeks without losing viability [97].

A highly interesting type of cell death was observed upon exposure of cells to sugars in specific conditions, such as in the absence of other nutrients in pure water [29,132,133,134], or as pulses in a carbon-source limited culture [135]. It is possible that these stimuli cause different types of death, because glucose solution was initially thought to cause apoptosis in *S. cerevisiae* [132], but was later shown to cause primary necrosis [29], while N-acetyl glucosamine in *C. albicans* caused a mixture of necrotic and apoptotic death [134]. The authors reporting on death due to the pulses of maltose in maltose limited culture hypothesized that death was due to the osmotic pressure caused by rapid maltose uptake [135].

Depletion of leucine has also been long known to cause death in leucine-auxotrophic yeast and the death was shown to be apoptotic as evidenced by *YCA1* dependence and annexin V staining [136]. The same paper showed that general depletion of nitrogen source can also cause cell death, but in this case the death is mostly necrotic. Notably, both these modes of cell death can be modulated towards increased or decreased death by deletion of different autophagy-related genes [136]; however, the authors of the mentioned study did not consider this cell death to be autophagic per se.

Depletion of inositol, a precursor of lipid synthesis as well as some signaling molecules, in cells that cannot synthesize it, causes rapid necrosis of *S. cerevisiae* (10-fold drop in viability in 7 h), but it could also be prevented by treatments preventing cell growth, such as cycloheximide. Thus, an apoptotic component cannot be excluded. The authors of the relevant paper proposed that the death was a result of imbalanced rates between the growth of the cell surface and the intracellular components [137]. Later work on *Schizosaccharomyces pombe* shows that inositol deprivation causes a mixture of apoptotic and necrotic death [138].

A large body of literature describes cell death in response to various other perturbations of lipid metabolism. As an in-depth review on the topic is available [139], we only wish to provide a general outline of the subject.

Yeast can consume fatty acids and integrate them into membrane phospholipids. However, increased concentrations of free fatty acids lead to necrotic cell death as evidenced by the loss of membrane integrity and the release of nuclear proteins [90]. Perturbations to the machinery for modification of fatty acids can also lead to their toxicity, as observed for cells lacking phosphatidic acid phosphatase [140] or those with combined ablation of the biosynthesis of phospholipids and triglycerides [141]. Treatment with polyunsaturated fatty acids also causes death via apoptotic and necrotic mechanisms [142,143], while a mono-unsaturated fatty acid causes necrotic death associated with accumulation of lipid droplets which is exacerbated in the absence of macromitophagy [57].

Sphingolipids are an important component of eukaryotic membranes and disruption of their synthesis via drugs such as aureobasidin A causes apoptotic cell death in budding yeast [144]. It is also possible that death due to the other perturbations also involves damage to sphingolipid homeostasis, because acetic acid treatment induces accumulation of ceramides [145]. C2-ceramide induces both caspase-independent apoptotic and necrotic cell death, accompanied by the release of ROS, most probably associated with the mitochondria [146]. C2-ceramide also disorganizes lipid rafts and leads to necrotic cell death under hypoosmotic conditions, which suggests a multiplicity of lipotoxic pathways [147]. Most of the commonly used antifungal drugs are also targeted at components of the lipid metabolism or interfere with lipid entities. Azole drugs, such as fluconazole, itraconazole ketoconazole, clotrimazole and others, are mainly thought to act via disruption of ergosterol synthesis and several of them can cause apoptosis-like cell death in [78,148] yeast. We must note that only one of these papers [149] assessed CFUs, and so it is possible that disruption of ergosterol synthesis might have provided false positive results of cell death, however this is speculative. Similar compounds are often used in agriculture (imazalil, oxpoconazole, etc.), but, unfortunately, death in response to these drugs has not been studied in detail.

Polyene drugs, typified by nystatin and amphotericin B, are widely used in medical practice, although the latter is used as a second-line treatment due to the fact of its toxicity. Interestingly, we found no detailed information on cell death caused by nystatin, in contrast to numerous papers using amphotericin B, and these show that it can cause both necrotic and apoptotic death. However, involvement of metacaspase has only been demonstrated in *C. albicans* [150,151].

Miltefosine is an analogue of phosphatidylcholine and inserts into cell membranes, potentially disrupting function of membrane-associated proteins. In *S. cerevisiae*, it disrupts membrane potential via an interaction with the *COX9*-encoded subunit VIIa of the COX complex of the electron transport chain and causes caspase-dependent apoptosis and necrosis [152,153]. In the *Cryptococcus* yeasts, miltefosine was reported to kill cells through interaction with ergosterol. It also affects the mitochondrial membrane, which ultimately leads to apoptosis [154].

The anticancer lysophospholipid edelfosin selectively modifies lipid rafts in the yeast plasma membrane, which leads to apoptotic death, accompanied by an increase in ROS levels and changes in the mitochondrial membrane potential [155,156].

Terbinafine inhibits the growth of *S. cerevisiae* by increasing the accumulation of squalene, which exhibits lipotoxic properties, accumulating in non-canonical droplet structures [157].

#### 3.3.4. Oxidants

Multiple oxidants have been tested and shown to cause cell death in yeast, but the mechanisms seem to differ. Compounds which seem to act through oxidation of proteins mostly cause necrosis [158,159]. On the other hand, hydrogen peroxide, which is widely used to induce apoptotic death in yeast at low concentrations, causes very little necrosis [160], at least at the tested incubation times. Interestingly, according to some reports, neither hydrogen peroxide nor menadione caused protein carbonylation [161], unlike allyl alcohol; however, another paper did report protein carbonylation induced by hydrogen peroxide [145,162]. Acetic acid, which also causes oxidative stress and triggers apoptosis, does not cause protein carbonylation [145].

Some metal-containing compounds are also thought to act by inducing oxidative stress, and our data analysis showed that most of the metals in our survey (cadmium, copper, chromium, and aluminum) caused accumulation of reactive oxygen species. Zinc was not tested for ROS formation in a cell death context, but genome wide screening for resistance to zinc did show that numerous mutants formed ROS in its presence [163]. Only manganese stress did not cause ROS formation, though in one paper [164] its toxicity was dependent on the mitochondria, which might implicate ROS.

Notably, metals mostly caused a mix of Yca1 dependent apoptotic and necrotic death, with the exception of zinc, which seems to cause death with characteristics of necrosis and autophagy [56] independent of Yca1.

Treatment of cells with metalloids and non-metals which are known for their oxidative behavior leads to caspase-dependent apoptosis. Arsenite increases the intracellular ROS and Ca^2+^ level [165], reduces the mitochondrial potential [166], but does not affect cells lacking the Tim18 mitochondrial translocase [167]. Selenium also induces oxidative stress, accompanied by genotoxic effects [168], and changes the composition of the plasma membrane, accompanied by morphological changes as assayed by electron microscopy [169]. With increasing exposure time, sodium selenite causes nuclear fragmentation and Aif1-dependent apoptosis [170]

#### 3.3.5. Acids

Weak acids, namely acetic acid, have been thoroughly studied for their effect on yeast cell death and these studies have been extensively reviewed [171,172]. They are thought to act mostly via acidification of the internal environment of the cells, since under low pH in the external medium, weak acids are undissociated and thus can permeate the PM. After entering cells, they dissociate and cause intracellular acidification. According to most reports, this causes both necrotic and apoptotic death. The latter can proceed via a yeast metacaspase Yca1-dependent pathway [173] or be Yca1-independent [174]. It has been suggested [175] that cell death due to the presence of several factors proceeds via intracellular acidification and generation of ROS, with the connection being that intracellular acidification results in the protonation of superoxide anion (O^2−^) to form the hydroperoxyl radical, one of the most aggressive ROS, which causes cell death. Stronger acids, such as HCl, also cause cell death via apoptotic and necrotic death, although it is unclear whether the necrosis is primary or secondary [176,177], and how much intracellular acidification these treatments actually cause. However, adaptation to acid stress can also be a mitigating factor, for instance, in the adaptation to acetic acid [178,179].

#### 3.3.6. Alcohols

Since one of the main uses of budding yeast is the production of ethanol and, more recently, other types of alcohol for biofuel, death in response to various alcohols has been studied by many researchers. Ethanol (and probably other alcohols) is thought to act on the cell membrane by inserting into the lipid bi-layer and perturbing its structure, at some point causing interdigitation between hydrophobic tails of phospholipids and consequent thinning of the membrane [180]. Most of the data we collected involved ethanol-mediated death (which has been reviewed separately, see [181]), and demonstrates a mixture of apoptotic and necrotic death. Somewhat contradictory, while caspase substrates seem to indicate caspase activation, deletion of the *YCA1* gene does not seem to have a mitigating effect. Isobutanol, a possible biofuel, can also kill yeast cells, and while it is unclear whether this alcohol induces mostly necrotic or apoptotic death, its effects can be reduced by expression of annexins, normally absent in *S. cerevisiae.* These proteins probably act by facilitating the repair of membrane damage [182].

#### 3.3.7. Antimicrobial Peptides

A range of peptides with antifungal properties have been studied in different species of yeast, with a considerable amount of work being done on *Candida* species (47 out of 75 papers). The available data indicate that peptides can induce different types of death, some of which involve non-permeabilizing apoptotic death (9 peptides) or a combination of apoptosis and necrosis (10 peptides). There are only four cases of exclusive necrosis, barring some possible involvement of autophagic/lysosomal death which were not tested. This low number is probably due to the fact that a large number of peptides that cause necrosis were not tested for hallmarks of apoptosis (30 peptides) (Appendix A). Overall, this is in accordance with the known mechanisms of action for antifungal peptides, which can either weaken cell walls, directly permeabilize membranes, or trigger pro-death signaling via the cell surface or by entering cells (reviewed in [183]).

Notably, for ~50% of peptides tested for apoptotic hallmarks, the death of cells seems to be accompanied by caspase activation (as judged by staining with FITC-VAD-FMK). Oxidative stress was also observed, although the causative role of ROS was not demonstrated in most cases.

#### 3.3.8. Additional Compounds Used in Medicine and Agriculture

The main medically relevant antifungal drugs currently in use are azoles, polyenes (described above) and echinocandins. Some of these groups and several other types of compounds are also used in agriculture (reviewed in [184]).

Echinocandin-like drugs, such as Caspofungin and Micafungin, target the yeast cell wall and specifically inhibit glucan synthase, thus triggering apoptosis and necrosis [185], suggesting that even though the overall picture of cell death involving ROS and apoptotic hallmarks seems similar to other stimuli, in this case the ROS originate from the ER and not the mitochondria. This was shown to involve the Ero1 protein which mediates formation of disulfide bonds during oxidative protein folding in the ER. Possibly during damage to the cell wall, the cell wall integrity signaling pathway increases production of secreted cell wall proteins, thus driving up Ero1-related oxidative stress.

Microtubule disrupting compounds are used in agricultural antifungal applications (benomyl), as anti-tumor drugs (paclitaxel), as well as in laboratory practice for cell cycle arrest (nocodazole). Naturally, due to the fact of their cell-cycle arresting capacity, these compounds stop cell division, but they also kill cells via apoptotic mechanisms after a certain amount of time. Notably, at least in the case of nocodazole, death is independent of metacaspase [186].

Several papers address the cell death caused by the antitumor drug cisplatin, which interacts with DNA and creates cross-links between DNA strands [187]. In yeast this drug causes apoptotic death without noticeable necrosis, accompanied by DNA-damage [39], which is quite expected, based on its mechanism of action reported from higher eukaryotes [188]. Interestingly, the is a report on the absence of ROS formation as assayed by several dyes, while in the absence of the *GSH1* gene, cells are sensitive to cisplatin and display increased lipid peroxidation both under normal and caloric restriction conditions, suggesting that oxidative stress might play a causative role in cisplatin-mediated cell death [39,189].

Many other compounds have been studied in relation to yeast cell death and they are difficult to group for systematization, thus we invite readers to peruse our compiled table (Appendix A).

## 4. Discussion

Our attempt to systematize the known data on cell death in response to different stimuli allows for several generalizations. The most obvious one, is that, even though most of the collected papers follow a highly similar experimental plan (schematically depicted in Figure 2), it is difficult to group the available data in such a manner that would allow direct comparison of the results. Mostly this is due to the somewhat arbitrary selection of conditions for assaying cell death. Thus, although we were unable to make direct comparisons between various data, our compilation allows selection of interesting stimuli, which can be compared to each other side by side in future experimental work to determine the necrotic and non-necrotic ranges of cell death for different stimuli.

Another general feature of the literature is that the only central governing paradigm for unifying the obtained data is the similarity of non-necrotic cell death in yeast and fungi to that of higher eukaryotes. We feel that a much wider range of possibilities should be considered by assaying how various factors (pH, various antioxidants, osmotic stabilization, caloric restriction, cycloheximide and other factors) can mitigate cell death. Clusterization analyses to identify groups of factors which demonstrate similar patterns of responses should provide valuable insight. Similar work should also be done with mutants harboring deletions of the genes known to modulate yeast cell death, in order to form clear groups of stimuli which trigger specific types of cell death. Perhaps progress in this and other areas will allow yeast researchers to transition from using the convenient, albeit imperfect parallels with cell death in higher eukaryotes to creating a novel and more adequate nomenclature and understanding for cell death in microorganisms, which will be based on a deeper knowledge of the molecular markers involved in death, as well as data on the involvement of specific cellular systems in the death process. As noted above, death in response to only ~20 stimuli have been studied in 5+ papers and, thus, the range of possible studies is vast.

On a more specific note, the causative role of commonly assayed factors, such as ROS and the activity of metacaspase, seem quite evident; however, our survey suggests that the apparent prevalence of their effects may be due to the fact of confirmation bias, and cases in which these factors show no causation or have seemingly not been tested can be re-examined in order to study novel mechanisms of cell death. There are also important methodological issues that highlight the importance of the specific setup of experiments and quality of data reporting in papers. Foremost, this involves the inclusion of raw flow cytometric data as well as analyzing cell permeability in parallel with most of the used assays such as FITC-VAD-FMK.

Another important feature is the need to determine criteria for denoting various stimuli as necrotic or apoptotic, such as by reporting necrosis/apoptosis percentages at specific treatment conditions, which can be easily determined and formalized across stimuli (such as MIC, MFC, LD_50_)

Our survey suggests that several areas of yeast cell death are undercharacterized but have the potential for fast progress or at least definitive clarification. Most obviously, these are the roles of lysosome- and autophagy-associated phenomena in cell death, because the study of autophagy in a physiological context has been most successful and provides numerous tools, but there is no clear understanding of the role or lack of role for lysosome (vacuole) permeabilization or autophagy in most stimuli causing cell death.

We observed a similar issue with calcium signaling. Although some work in this direction has been done and parallels with higher eukaryote cell death suggest that calcium signaling should play an important role in yeast cell death, this area is much less well characterized than the role of mitochondria and ROS. Gathering wider knowledge on the role of calcium in yeast cell death is all the more important because Yca1 has been shown to be directly activated by calcium ions [190]. A schematic representation of our concluding recommendations for further study of cell death in yeast is presented in Figure 6.

This review is, to our knowledge, the first attempt to systematize the vast amount of data on cell death induced by various factors in yeast and other fungi. We are sure that some papers reporting on cell death and some of its aspects in yeast have been missed, and also that more have been published during the writing of this paper and will be published after. We invite authors that would like their papers included in our compilation or who would like to add data on existing papers, to contact us, so that in a joint effort we can increase the quality of the resource we have created. To make data compilation easier, we have created a form for entering data that can be accessed at shorturl.at/huxG5. The online version of our table, which we plan to update when possible, is available at shorturl.at/lBU07. Our nearest plans are to provide systematic data on the analysis of cell death induced by internal changes, such as mutations and expression of various proteins, which were excluded from this paper. We are also interested in creating approaches to more rapidly compile the data and increase the specificity of our searching strategies, because gathering systematic data of work on cell death in yeast is non-trivial and is not easily covered by available search strategies. Any feedback in these directions would be appreciated.

## Figures and Tables

**Figure 1 jof-07-00886-f001:**
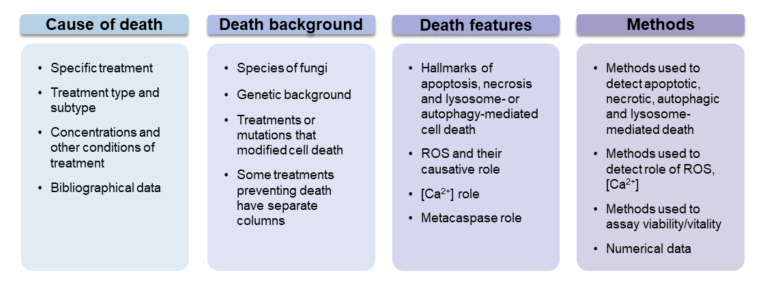
Overview of data associated with each cell death stimulus from each paper as compiled in Appendix A.

**Figure 2 jof-07-00886-f002:**
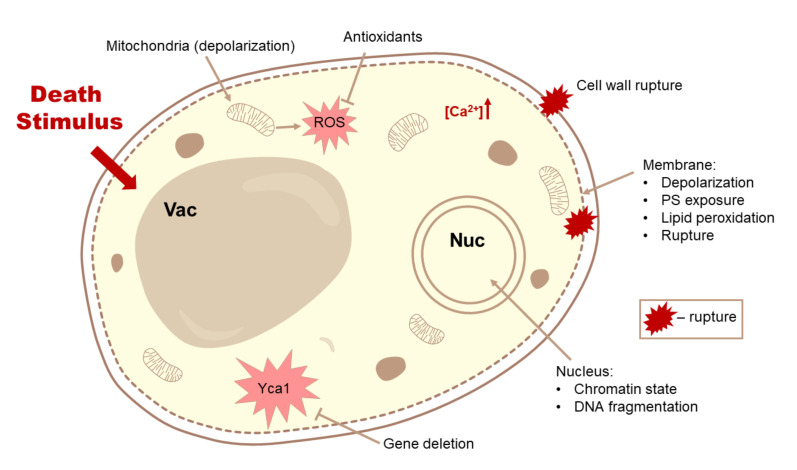
Commonly assayed features associated with cell death in either a correlative or causative manner.

**Figure 3 jof-07-00886-f003:**
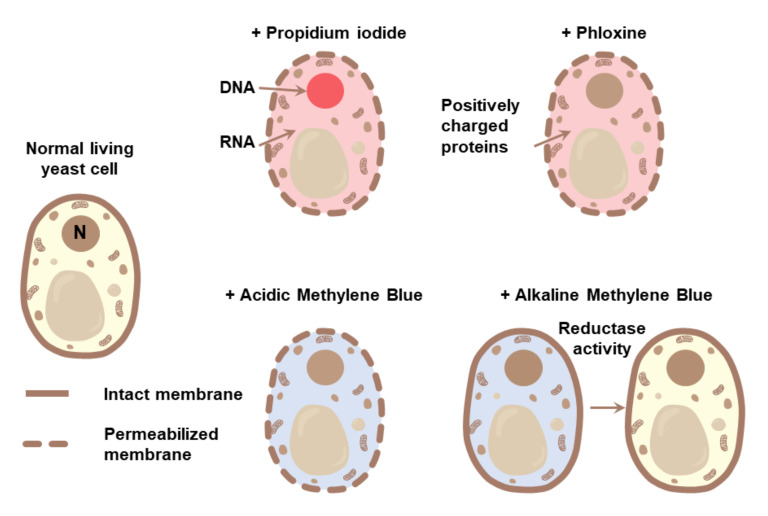
Schematic of the action of the methylene blue and phloxine B membrane permeability dyes.

**Figure 4 jof-07-00886-f004:**
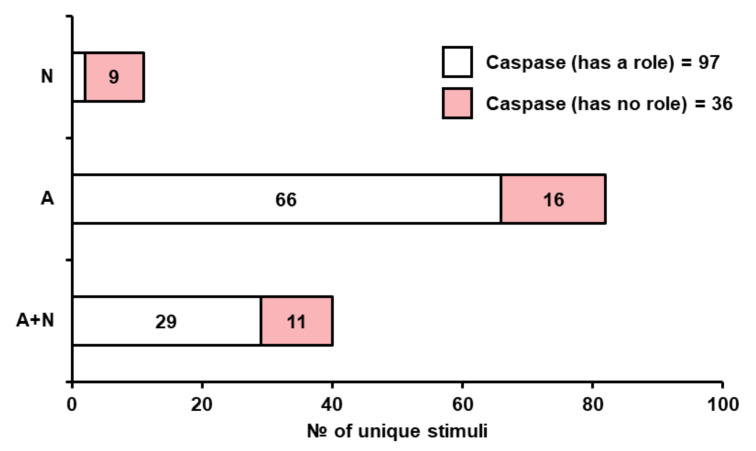
Role of Yca1 or its analog in cell death caused by various stimuli. A + N—apoptosis and necrosis hallmarks observed together; A—exclusive apoptosis; N—exclusive necrosis.

**Figure 5 jof-07-00886-f005:**
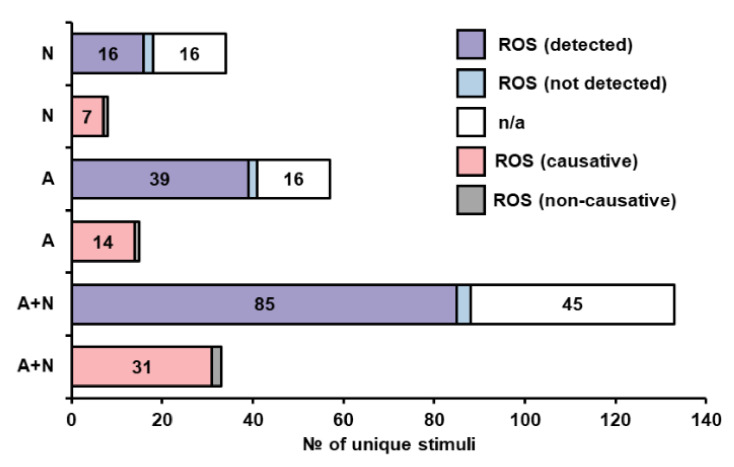
ROS generation and its role in cell death caused by various stimuli. Generation of reactive oxygen species are common features of cell death in yeast cells, whether it is necrotic, apoptotic, or mixed death. A + N—Apoptosis and necrosis hallmarks observed together; A—exclusive apoptosis; N—exclusive necrosis.

**Figure 6 jof-07-00886-f006:**
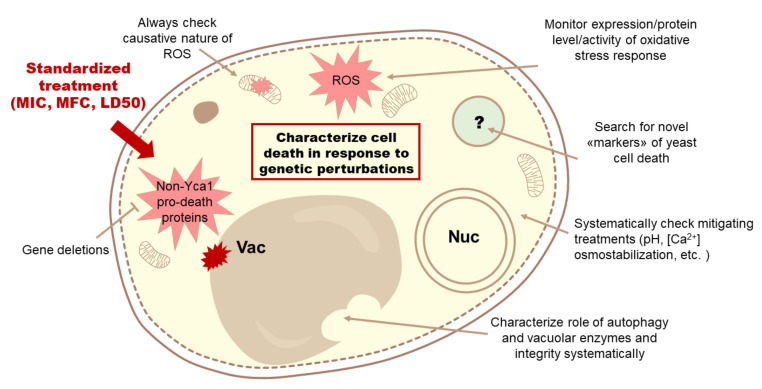
Recommendations for improving future cell death studies in yeast.

## Data Availability

Not applicable.

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
