# Peer review of "A Systematic Survey of Characteristic Features of Yeast Cell Death Triggered by External Factors"

_jof, 2021, doi:10.3390/jof7110886_

Round 1
Reviewer 1 Report
The manuscript by Grosfeld et al. is a broad and critical survey comparing cell death induced by different external stimuli in different yeast, such as Saccharomyces cerevisiae, Schizosaccharomyces pombe, Candida albicans. Authors focus on methodologies for cell death detection and different death inducing perturbations, such as physiological stimuli or physical/chemical treatments. Less attention is paid to the mechanisms of cell death, as the same authors state. Some proper criticisms are raised on the issue regarding the severity and duration of death-inducing stimuli and their result in different types of death. At this regard, authors propose to create formal criteria which could help to classify cell death type and stimulus intensity with cell death markers.
The manuscript is a well written attempt to systematize the vast amount of data on cell death induced by various factors in yeast and provides interesting suggestions from different perspectives. However several points require a more analytical description, integration with the existing literature and discussion and need to be revised.
Here follow my comments:
Paragraph 3.3.5.
Fig.1, panel on Death background: change "modified" with another verb, such as"influence or modulate"
Legend of Fig.2 is a discussion on membrane permeability dyes. It would be useful to include these informations in the main text and explain just the figure in the legend.
Legend of Fig.3. The title does not reflect entirely the content of the figure. I would suggest a different title: such as ROS generation and its role in cell death caused by various stimuli.
Line 410 add production/activities of proteins involved in the oxidative stress response
Lines 408-412 Studies related to these aspects are available in S.cerevisiae, thus the following articles should be cited in the main text: PMID 15894436; PMID: 30931079
Line 475, the title of this paragraph is not clear
Paragraph 3.3, lines 495-6
It could be useful to anticipate the criteria of classification of death-inducing stimuli assembling in physiological, physical, nutritionals, chemicals etc….
Lines 502, 516, 528, 542 etc… may be bold
Line 758-760: This sentence is not clear. What causes both apoptotic and necrotic cell death? If the authors refer to acetic acid-induced regulated cell death, which can be -dependent or -independent on the yeast metacaspase YCA1, this can be exclusively apoptotic occuring via two different pathways.
The title of the manuscript remains too generic. I would suggest to better highlight the scope of the work.
Check in the whole manuscript the positions of references in the text (for example: lines 819-823 move the citation numbers at the end of the sentence).
Check whether the title of subparagraphs has to be bold or not.
Author Response
- Fig.1, panel on Death background: change "modified" with another verb, such as"influence or modulate" - Changed
- Legend of Fig.2 is a discussion on membrane permeability dyes. It would be useful to include these informations in the main text and explain just the figure in the legend. - We have moved this text into the manuscript
- Legend of Fig.3. The title does not reflect entirely the content of the figure. I would suggest a different title: such as ROS generation and its role in cell death caused by various stimuli. - We have changed the phrasing according to the reviewers suggestion
- Line 410 add production/activities of proteins involved in the oxidative stress response - Changed
- Lines 408-412 Studies related to these aspects are available in S.cerevisiae, thus the following articles should be cited in the main text: PMID 15894436; PMID: 30931079 - We have added these references
- Line 475, the title of this paragraph is not clear - Сhanged to “Role of treatment severity in determining cell death type”
- Paragraph 3.3, lines 495-6 It could be useful to anticipate the criteria of classification of death-inducing stimuli assembling in physiological, physical, nutritionals, chemicals etc…. - We have done this
- Lines 502, 516, 528, 542 etc… may be bold - Changed
- Line 758-760: This sentence is not clear. What causes both apoptotic and necrotic cell death? If the authors refer to acetic acid-induced regulated cell death, which can be -dependent or -independent on the yeast metacaspase YCA1, this can be exclusively apoptotic occuring via two different pathways. - Changed
- The title of the manuscript remains too generic. I would suggest to better highlight the scope of the work. - We have modified the title to “A systematic census of characteristic features of yeast cell death triggered by external factors”. We thank the reviewer for suggesting a change to the title and hope that this the current one is more eye catching and representative of the unique aims of this work
- Check in the whole manuscript the positions of references in the text (for example: lines 819-823 move the citation numbers at the end of the sentence). - We have checked and moved the references where appropriate
- Check whether the title of subparagraphs has to be bold or not. - This formatting issue is probably better decided by the technical editors
Reviewer 2 Report
In this manuscript, the authors summarize recent progress in the research field of yeast cell death, which is receiving increasing attention. They perform a careful and through analysis of the mechanisms of cell death, and provide novel markers useful for discrimination of the types of cell death. This study could help our understanding of the mechanism and physiological relevance of yeast cell death.
Author Response
We thank the Reviewer for their positive comment and careful review.
Reviewer 3 Report
The review paper by Grosfeld and coworkers represents an interesting effort to systematize the triggers, characteristics and investigation methods of yeast cell death caused by external stimuli. It provides a huge amount of information and it is generally well written. It will be useful to both investigators already in the filed as well as newcomers. I’m summarizing below a few points that could improve the paper.
1.- Please, homogenize the different supplementary tables in terms of format and uniformity style. For instance:
- homogenize size and color of font in Table S1, tab “Abbreviations and Methods” and adapt row height to text.
- Define heading in column A (blank in many of the tabs).
- Avoid uninformative tab label (such as “sheet4” in Table S4).
- Avoid inconsistencies between text and tables (line 388, table S6, tab X, but no tab X is found.)
- Remove “copy of” in tabs of Table S6,…
2.- The review is a bit short of figures. It could benefit of translating some key information from the text into extra graphs or cartoons. Note also that the use of in figure 3 colors (shadows of blue and grey) is far from optimal. Please, use colors that are easier to distinguish.
3.- Albeit writing is generally clear and good, please check for possible improvements (i.e. line 67-68, “condensation of chromatin, fragmentation of DNA, (AND) exposure of phosphatidylserine.).
4.- Finally, have the authors considered stablishing a searchable WEB-based database containing the information present in the different Supp. Tables and able to collect data supplied by users? Such tool would be very useful and, in addition, would increase the number of citations of the paper.
Author Response
1.- Please, homogenize the different supplementary tables in terms of format and uniformity style. - We have made the requested corrections and tried to remove additional inconsistencies throught the paper
2.- The review is a bit short of figures. It could benefit of translating some key information from the text into extra graphs or cartoons. Note also that the use of in figure 3 colors (shadows of blue and grey) is far from optimal. Please, use colors that are easier to distinguish. - We have added several figures which highlight our conclusions, as well as changed the color scheme of some of the previously present figures.
3.- Albeit writing is generally clear and good, please check for possible improvements (i.e. line 67-68, “condensation of chromatin, fragmentation of DNA, (AND) exposure of phosphatidylserine.). - Changed
4.- Finally, have the authors considered stablishing a searchable WEB-based database containing the information present in the different Supp. Tables and able to collect data supplied by users? Such tool would be very useful and, in addition, would increase the number of citations of the paper. - As such the file we supply is searchable inside, while we do not see how this tool could be searchable via search engines (if we understand the reviewer correctly). An upcoming review that we are currently working on, will add data on genetic perturbations that cause cell death, and thus we can expand and organize the material during the next step of this work. As of now, we include an online version of the table that we will update, as well as an online form for adding new papers to the “database”, if other authors will wish to do so (see final paragraph).
Reviewer 4 Report
This manuscript compiles numerous papers on the issue of cell death in yeast. The amount of data is overwhelming. In general, it is well written but sometimes the information is difficult to digests and the paragraphs are too long. Revision of the treatments that cause cell dead in yeast is exhaustive. The authors have done a very good job posing the pros and the contras of each type of treatment, however, sometimes is not easy to obtain a conclusion. Overall, this is a well-performed study and addressing my following concerns may improve the manuscript.
Minor comments
1:- The figures are small and scarce, thus my suggestion is that the paper will be more attractive with more figures. For instance, you can include a figure or scheme showing the differences of yeast cells dying for necrosis, for apoptosis-like death and for necrosis plus apoptosis.
2.- line 24. I have problems to understand this long sentence “The table presents a resource for orientation within the literature and suggests that study of yeast cell death has been driven by a paradigm of similarity between non-necrotic death in yeast and apoptosis in mammals” …the word suggests means that this is a conclusion inferred from this study? … I will propose to use model instead of paradigm….
3.- Line 125. In total, we included data collected from the full texts of 691 papers, with data on 686 unique stimuli…. Here, unique is synonymous of distinctive, different? Please clarify that point.
4.- line 168. Where is Box1???
- Line 188. “Among high throughput methods, luciferase-based detection of adenylate kinase leakage seems to be a highly convenient, yet seldom used method [35,36], however the papers that report its use do not provide an easy way of determining the share of permeable cells, thus it is not clear whether this method can be used for quantitative comparisons between different stimuli, since permeabilization via different mechanisms might release protein with different efficiency”.
Very long sentence….and the question is, the luciferase-based detection of adenylate kinase leakage is or is not a very good method?
I suggest to shorten the paragraph as follows: Among high throughput methods, luciferase-based detection of adenylate kinase leakage seems to be a highly convenient, yet seldom used method [35,36], however this method cannot be used for quantitative comparisons because the papers that report its use do not provide an easy way of determining the share of permeable cells.
6.- Line 197. FUN1, as well as FDA. What FUN1 and FDA stands for?
7.- Line 353…At least some data on the effects of autophagy or vacuolar proteolysis on cell death are available for 37 stimuli, with indication of changes to the lysosomal activity or autophagy available for 26 stimuli, however in most cases cases, except for one [56], involving zinc toxicity, perturbation of the mentioned genes increased sensitivity to the death-stimulus. Very long sentence.
8.- Line 356… ever in most cases cases…delete one cases.
9.- Line 370…(Perrone et al., 2008)..should be re-numbered.
10.- Line 490 ..”For instance, percentages of necrotic and apoptotic cells at MIC, 50% lethality concentrations and MFC (with identical treatment times and medium) could be measured and compared for different stimuli”. I wonder if the sentence is correct or “and MFC should be changed for at MFC.
11.- Line 557. However, inhibition of this type of death uncovered a slower wave of death with different properties [72]. As indicated in reference 72, a slower wave of death must be changed to several waves of death.
12.- Line 558…in the text “Interestingly, there are reports that pheromone-induced death seems to be un-conserved among different strains of S. cerevisiae [99], while it can be observed in Candida albicans [73]”.
The word observed should be conserved. In the abstract of reference 73 …textually, the authors state, “we demonstrate that levels of PID vary widely between clinical isolates of C. albicans, with some strains experiencing close to 70% cell death”.
13.- Regarding death via interaction among yeasts species. It is known that some saccharomyces are able to secrete a number of toxic proteins “the killer toxin” which are lethal to susceptible cells. The effect of the killer toxin has not been addressed in the compiled material. Do you know what kind of cell death induces the killer toxin? Are there reports in the literature?
- Line 617…”Both of these cause noticeable permeabilization of the membrane [114,115] , but the cell death is not completely explained by cell permeabilization in one of the reports [115], supporting the results of [113] which indicate”. The last sentence is confusing, which indicate what? It could be supporting the results indicated in [113]?
-
15.- Line 700.. (Kavakçıoğlu and Tarhan, 2018; Lee and Lee, 2018a) should be corrected.
16.- Supplemental Methods Section for Data Compilation and Analysis.
What .py file mean??
…."pmid"…should be pmdi
Author Response
1:- The figures are small and scarce, thus my suggestion is that the paper will be more attractive with more figures. For instance, you can include a figure or scheme showing the differences of yeast cells dying for necrosis, for apoptosis-like death and for necrosis plus apoptosis. - We have added several figures to highlight the current state of the field, as well as a similar figure to highlight our recommendations for further studies. We have also modified the color scheme of some figures for increase ease of viewing. ((We have also added additional graphs to illustrate numerical data))
2.- line 24. I have problems to understand this long sentence “The table presents a resource for orientation within the literature and suggests that study of yeast cell death has been driven by a paradigm of similarity between non-necrotic death in yeast and apoptosis in mammals” …the word suggests means that this is a conclusion inferred from this study? … I will propose to use model instead of paradigm…. - We have corrected the abstract to more clearly represent our views on the results of the work. We have used the word view instead of model or paradigm
3.- Line 125. In total, we included data collected from the full texts of 691 papers, with data on 686 unique stimuli…. Here, unique is synonymous of distinctive, different? Please clarify that point. - We have changed the wording to different
4.- line 168. Where is Box1??? - Changed
5. Line 188.
Very long sentence….and the question is, the luciferase-based detection of adenylate kinase leakage is or is not a very good method?
- We have reworded this paragraph with shorter and clearer sentences
6.- Line 197. FUN1, as well as FDA. What FUN1 and FDA stands for?
- We have added the full-length names of the abbreviated substances
7.- Line 353…Very long sentence.
- We have split this sentence into two
8.- Line 356… ever in most cases cases…delete one cases.
- done
9.- Line 370…(Perrone et al., 2008)..should be re-numbered.
- done
10.- Line 490 ..”For instance, percentages of necrotic and apoptotic cells at MIC, 50% lethality concentrations and MFC (with identical treatment times and medium) could be measured and compared for different stimuli”. I wonder if the sentence is correct or “and MFC should be changed for at MFC.
- We feel that since this is a enumeration of several concentrations, only one “at” is sufficient
11.- Line 557. However, inhibition of this type of death uncovered a slower wave of death with different properties [72]. As indicated in reference 72, a slower wave of death must be changed to several waves of death.
- Corrected to multiple slower waves
12.- Line 558…in the text “Interestingly, there are reports that pheromone-induced death seems to be un-conserved among different strains of S. cerevisiae [99], while it can be observed in Candida albicans [73]”.
The word observed should be conserved. In the abstract of reference 73 …textually, the authors state, “we demonstrate that levels of PID vary widely between clinical isolates of C. albicans, with some strains experiencing close to 70% cell death”.
- After reviewing both papers, we have reworded this phrase: Interestingly, there are reports that pheromone-induced death seems to be variable both among different strains of S. cerevisiae [100], as well as isolates of Candida albicans [74]. Thus, although this type of death is present in different yeast genera, it does not seem to be highly conserved.
We feel this statement more adequately reflects the reported data.
13.- Regarding death via interaction among yeasts species. It is known that some saccharomyces are able to secrete a number of toxic proteins “the killer toxin” which are lethal to susceptible cells. The effect of the killer toxin has not been addressed in the compiled material. Do you know what kind of cell death induces the killer toxin? Are there reports in the literature?
- This is an excellent use case for our compiled material. We have reviewed our list and we have 11 papers concerning cell death caused by various killer toxins. We have written an additional section about this type of cell death.
- Line 617…”Both of these cause noticeable permeabilization of the membrane [114,115] , but the cell death is not completely explained by cell permeabilization in one of the reports [115], supporting the results of [113] which indicate”. The last sentence is confusing, which indicate what? It could be supporting the results indicated in [113] - We have reworded this section
- Line 700.. (Kavakçıoğlu and Tarhan, 2018; Lee and Lee, 2018a) should be corrected. - done
16.- Supplemental Methods Section for Data Compilation and Analysis.
What .py file mean??
…."pmid"…should be pmdi
- The .py file is an extension of Python-script files, we have added this information. PMID is pubmed ID, so this abbreviation is correct.
Round 2
Reviewer 1 Report
Only a minor point, I would suggest an alternative title for this paper:
Systematic survey of yeast cell death features triggered by external factors.
The authors have addressed all the raised points.
Author Response
Done. We thank the Reviewer for their positive comment and careful review.